# Estimates and Determinants of SARS-Cov-2 Seroprevalence and Infection Fatality Ratio Using Latent Class Analysis: The Population-Based Tirschenreuth Study in the Hardest-Hit German County in Spring 2020

**DOI:** 10.3390/v13061118

**Published:** 2021-06-10

**Authors:** Ralf Wagner, David Peterhoff, Stephanie Beileke, Felix Günther, Melanie Berr, Sebastian Einhauser, Anja Schütz, Hans Helmut Niller, Philipp Steininger, Antje Knöll, Matthias Tenbusch, Clara Maier, Klaus Korn, Klaus J. Stark, André Gessner, Ralph Burkhardt, Michael Kabesch, Holger Schedl, Helmut Küchenhoff, Annette B. Pfahlberg, Iris M. Heid, Olaf Gefeller, Klaus Überla

**Affiliations:** 1Institute of Medical Microbiology and Hygiene, Molecular Microbiology (Virology), University of Regensburg, Franz-Josef-Strauß-Allee 11, 93053 Regensburg, Germany; david.peterhoff@klinik.uni-regensburg.de (D.P.); melanie.berr@klinik.uni-regensburg.de (M.B.); sebastian.einhauser@klinik.uni-regensburg.de (S.E.); anja.schuetz@klinik.uni-regensburg.de (A.S.); Hans-Helmut.Niller@klinik.uni-regensburg.de (H.H.N.); Andre.Gessner@klinik.uni-regensburg.de (A.G.); 2Institute of Clinical Microbiology and Hygiene, University Hospital Regensburg, Franz-Josef-Strauß-Allee 11, 93053 Regensburg, Germany; 3Institute of Clinical and Molecular Virology, University Hospital Erlangen, Friedrich-Alexander Universität Erlangen-Nürnberg, Schlossgarten 4, 91054 Erlangen, Germany; Stephanie.Beileke@uk-erlangen.de (S.B.); philipp.steininger@uk-erlangen.de (P.S.); antje.knoell@uk-erlangen.de (A.K.); matthias.tenbusch@fau.de (M.T.); Clara.maier@extern.uk-erlangen.de (C.M.); Klaus.Korn@uk-erlangen.de (K.K.); 4Statistical Consulting Unit StaBLab, Department of Statistics, LMU Munich, Geschwister-Scholl-Platz 1, 80539 Munich, Germany; felix.guenther@stat.uni-muenchen.de (F.G.); kuechenhoff@stat.uni-muenchen.de (H.K.); 5Department of Genetic Epidemiology, University of Regensburg, Franz-Josef-Strauß-Allee 11, 93053 Regensburg, Germany; Klaus.stark@ur.de (K.J.S.); Iris.heid@ur.de (I.M.H.); 6Institute of Clinical Chemistry and Laboratory Medicine, University Hospital Regensburg, Franz-Josef-Strauß-Allee 11, 93053 Regensburg, Germany; ralph.burkhardt@klinik.uni-regensburg.de; 7University Children’s Hospital Regensburg (KUNO) at the Hospital St. Hedwig of the Order of St. John, University of Regensburg, Steinmetzstraße 1-3, 93049 Regensburg, Germany; Michael.Kabesch@barmherzige-regensburg.de; 8Bayerisches Rotes Kreuz, Kreisverband Tirschenreuth, Egerstraße 21, 95643 Tirschenreuth, Germany; schedl@kvtirschenreuth.brk.de; 9Department of Medical Informatics, Biometry and Epidemiology, Friedrich-Alexander Universität Erlangen-Nürnberg (FAU), Waldstr. 6, 91054 Erlangen, Germany; annette.pfahlberg@fau.de (A.B.P.); olaf.gefeller@fau.de (O.G.)

**Keywords:** SARS-CoV-2, seroprevalence, ELISA, CLIA, latent class analysis, antibodies, infection fatality ratio, underreported infections, smoking, senior care homes

## Abstract

SARS-CoV-2 infection fatality ratios (IFR) remain controversially discussed with implications for political measures. The German county of Tirschenreuth suffered a severe SARS-CoV-2 outbreak in spring 2020, with particularly high case fatality ratio (CFR). To estimate seroprevalence, underreported infections, and IFR for the Tirschenreuth population aged ≥14 years in June/July 2020, we conducted a population-based study including home visits for the elderly, and analyzed 4203 participants for SARS-CoV-2 antibodies via three antibody tests. Latent class analysis yielded 8.6% standardized county-wide seroprevalence, a factor of underreported infections of 5.0, and 2.5% overall IFR. Seroprevalence was two-fold higher among medical workers and one third among current smokers with similar proportions of registered infections. While seroprevalence did not show an age-trend, the factor of underreported infections was 12.2 in the young versus 1.7 for ≥85-year-old. Age-specific IFRs were <0.5% below 60 years of age, 1.0% for age 60–69, and 13.2% for age 70+. Senior care homes accounted for 45% of COVID-19-related deaths, reflected by an IFR of 7.5% among individuals aged 70+ and an overall IFR of 1.4% when excluding senior care home residents from our computation. Our data underscore senior care home infections as key determinant of IFR additionally to age, insufficient targeted testing in the young, and the need for further investigations on behavioral or molecular causes of the fewer infections among current smokers.

## 1. Introduction

COVID-19 case numbers reported to health authorities based on PCR testing continue to rise worldwide, but the precise cumulative number of infected individuals remains unknown. PCR testing frequencies and strategies vary largely between countries and over time, thus limiting the strength of the conclusions that can be drawn based on case fatality ratios (ratio of SARS-CoV-2 related deaths to the number of PCR positive cases reported to health authorities, CFR) [1]. Determining the number of infected individuals, the ratio of underreported SARS-CoV-2 infections, and the ratio of the number of COVID-19 related deaths to the number of infected (infection-fatality ratio, IFR) helps to understand the extent of undetected infections (factor of underreported infections), to determine the level of herd immunity, and to instruct public health measures. The gold standard for assessing the cumulative case numbers of viral infections are population-based seroprevalence studies within an appropriate time-period after outbreak based on random sampling from public registries. At low seroprevalence, even a small deviation from 100% specificity of the tests used for determining antibody responses to SARS-CoV-2 can lead to bias due to a low positive predictive value. Additional adjustments considering the decay of antibody levels after infection may also be necessary, although recent results indicate the stability of IgG levels against the SARS-CoV-2 spike protein for more than six months [2].

In Europe, Italy was the first country to be hit hard with more than one confirmed COVID-19 case/100,000 inhabitants/14 days nationwide on 28 February 2020. With a delay of one to two weeks, case counts in other European countries such as Spain, France, Germany, the UK, and Portugal passed this level [3,4]. In Germany, a first cluster of COVID-19 cases occurred between 27 January to 19 February 2020, but was contained by contact tracing [5,6]. Most likely, ski-travelers returning from Austria and Italy and the carnival festivities were important determinants of the subsequent initial spread of SARS-CoV-2 in Germany and the number of confirmed cases exceeded 1000 by 10 March. The consequence of a superspreading event during carnival on 15 February in Gangelt, a municipality in the county of Heinsberg in North Rhine Westphalia, was analyzed by a household-based, seroprevalence study. While 3.1% of the population had been reported SARS-CoV-2 positive by PCR at the time of the study, the seroprevalence at this time in Gangelt was 14.11%. Based on only seven early deaths of COVID-19 cases reported until 6 April in Gangelt, this resulted in an inferred CFR of 1.8% and an inferred IFR of 0.36% [7].

A second hotspot of COVID-19 cases in Germany occurred in the county of Tirschenreuth located in the northeast of Bavaria with the first recognized COVID-19 case on 17 February 2020. Daily case counts peaked at 55 on 16 March and a stay-at-home order was issued for the hardest hit municipality in that county on 18 March. Until 11 May, the incidence of new COVID-19 cases declined to sporadic cases. By then, the number of total cases had summed up to 1122 corresponding to 1548 cases/100,000 inhabitants [8,9], the highest incidence observed in Germany for any county until November 2020. In addition, an extraordinary high case-fatality ratio (CFR, 11.5%) prompted an epidemiological investigation by the Robert Koch Institute, Germany´s federal center for infectious disease control. This investigation concluded that travelers returning from skiing vacations in Italy or Austria prior to detection of the first case in Tirschenreuth, early undetected community transmissions, and ultimately a beer festival contributed to the steep increase in COVID-19 cases starting 10 March, but were not sufficient to explain it entirely [8,9].

One explanation for the high CFR could be an increased infection occurrence in the most vulnerable age groups: median age of COVID-19 cases in the county of Tirschenreuth was 56 years compared to Germany and Bavaria (50 years each) or Gangelt (52 years) and the percentage of COVID-19 cases aged 70 years or older among all COVID cases was higher in Tirschenreuth than in the rest of Germany [9]. In addition, 56 of the 129 deaths observed in Tirschenreuth until 11 May occurred in senior care homes, resulting in a CFRs of 36% for residents of senior care homes and 7.5% for inhabitants of Tirschenreuth not living in care homes [9]. However, both CFRs were higher than the ones observed for residents of care homes for the elderly (20%) or the general population (4.4%) in Germany during the same time period. In addition, the CFR in the general population in the county of Tirschenreuth was more than 4-fold higher than the CFR initially reported for the municipality of Gangelt [7].

The small percentage of asymptomatic cases among the registered cases in Tirschenreuth [9] further suggests that a larger proportion of asymptomatic SARS-CoV-2 infections remained undetected, particularly at this very early phase of the pandemic in Europe. Since differences in testing frequencies and strategies may affect the percentage of reported SARS-CoV-2 infections and thus the CFR, the determination of the seroprevalence rate in the Tirschenreuth population will allow us to calculate the IFR and help to better understand the discrepant observations regarding the CFR.

The IFR is an important parameter to predict the risk that health care systems are overwhelmed by the COVID-19 pandemic. Many political decisions on contact reduction measures are still made without a good knowledge of the precise IFR for the targeted population. A recent meta-analysis also revealed that the IFR can vary substantially across different locations [10], possibly reflecting differences in age and risk structure among others.

To determine the seroprevalence, the factor of underreported infections and the IFR in the county of Tirschenreuth, we determined the prevalence of antibodies against SARS-CoV-2 in a population-based study based on a random sampling approach and three independent SARS-CoV-2 antibody tests. Demographic and lifestyle factors potentially associated with seroprevalence were also assessed by a questionnaire.

## 2. Subjects and Methods

### 2.1. Study Design

The county of Tirschenreuth, located in the northeastern part of Bavaria immediately adjacent to the Czech Republic, is known as one of the hot spots of SARS-CoV2 spread in Germany during spring and early summer 2020 with —until November 2020—the highest percentage in PCR diagnosed infections/100,000 inhabitants all over Germany until November 2020.

TiKoCo was designed as prospective cohort study with a baseline survey to determine the SARS-CoV-2 seroprevalence in the county of Tirschenreuth and two follow-up investigations four and nine months after baseline to monitor its temporal development, focusing on the durability of antibody responses, incidence of new infections, and potential secondary infections. Further primary goals of this baseline survey were to quantify the percentage of undetected and therefore underreported infections and to determine the infection fatality ratio (IFR) in the overall population and depending on age, sex, and residency in the various municipalities of the county Tirschenreuth. We focused on yielding robust data on seroprevalence by determining serum antibodies directed against two different viral antigens, the SARS-CoV-2 nucleoprotein as well as the viral spike protein (S) and the S protein receptor-binding domain (RBD) by three independent assays. Further aspects of interest were (i) the association of the individuals’ report of COVID-19 related symptoms and antibody status, particularly among these who were not aware of their infection (i.e., excluding individuals with a report of positive PCR-test) and (ii) the association of education, individual household situation, and lifestyle factors with antibody status.

The TiKoCo study was approved by the Ethics Committee of the University of Regensburg, Germany (vote 12-101-0258) and adopted by the Ethics Committee of the University of Erlangen (vote 248_20 Bc). The study complies with the 1964 Declaration of Helsinki and its later amendments. All participants provided written informed consent.

### 2.2. Study Capture Area and Eligible Individuals

The county of Tirschenreuth captured 64,643 aged 14 years or older inhabitants of mostly Caucasian ethnicity including 12,066 aged 70+. The county comprises 26 municipalities with populations varying from 822 to 7807, respectively. Eligibility criteria for participation was the willingness to come to the indicated study center and to provide 5.7 mL blood. Inclusion criteria were legal age and univalent consent of custodial parents for those aged <18 years, German language skills sufficiently good to understand the participant information, and to respond to the questionnaire. Children below 14 years and individuals with legal guardian were excluded from participation.

### 2.3. Sample Size

Assuming a SARS-CoV-2 seroprevalence of 10% in the population of Tirschenreuth, the participation of 3600 individuals would result in 95% confidence intervals with a width of ±1%, which was considered satisfactory. With regard to a future longitudinal survey, changes in seroprevalence between two sampling time points of 0.1% could also be proven statistically significant at a predefined statistical power of 80%. We were conservatively assuming a response rate of 50% or above for our study. This suggested that up to 7200 randomly sampled individuals were to be contacted to successfully recruit the calculated 3600 baseline study participants.

### 2.4. Study Participant Recruitment

For the baseline survey, we selected a representative sample of 7200 residents of Tirschenreuth aged 14 years and older by means of a sex- and municipality-stratified random sample. Based on data on county residents regarding sex and age, which were available for each municipality (Bayerisches Landesamt für Statistik 2019), we determined, for each of the 26 municipalities, the number of men and women, which corresponded to the share of the members of the municipality within the total population of Tirschenreuth, respectively.

The various municipalities in Tirschenreuth were approached with specific target figures and asked to randomly sample men and women aged 14 years or older (as per 2 June), collectively adding up to a total sample of 7200 potential study participants (Figure 1). Between 17 June and 26 June 2020, we mailed invitation letters to two thirds of these pre-selected individuals with the request to participate in a prospective COVID-19 antibody study. Invitations included written information regarding the study plan and goals, participant information, and a questionnaire (see below). Depending on their home address, invited individuals were asked to visit the nearest of the three study centers for a given date between 28 June and 10 July to donate 5.7 mL blood, to deliver the filled-in questionnaire, receive the data protection declaration, obtain personal advice from study staff if requested, and sign the declaration of consent. A total of 104 handicapped or otherwise immobile individuals were visited on their request at home for the blood draw. Invited individuals with flu-like symptoms were asked to stay at home and use the installed hotline to agree on a date for a visit at home for blood draw, additionally offering a nasal throat swab. Changing the date of and location for the blood sampling via the hotline was optional. After the second day of blood sampling (29 June 2020), the participation ratio based on the 4801 initially contacted persons was calculated to decide on the number of individuals for the second batch of invitations. This resulted in another 1807 individuals who were contacted via mail and asked to visit one of the test centers in the week of the 13th of July. Altogether, 6540 individuals were contactable.

### 2.5. Measures to Minimize and Understand Non-Response

Individuals who showed up at the study center at the dates as communicated in the invitation letter or as otherwise agreed via the hotline were considered immediate responders. A reminder invitation letter was sent out to the 1403 initial non-responders who were re-invited for July 13 to July 17. Those who responded following this reminder were considered late responders (*n* = 265). Of the 6608 sent out invitation letters, 68 were returned (not contactable individuals). It turned out that the residents at 10 of these addressees had died (reason unknown) and one was hospitalized due to COVID-19. Individuals who received the invitation and did not visit the study center or asked for an alternative appointment, not even after the invitation reminder, were considered as non-responders.

### 2.6. Questionnaire

Together with the invitation mailed to individuals eligible for our study, we sent out a self-administered questionnaire and asked individuals to bring the filled-out questionnaire to the study center offering personal counseling by trained staff members in case of questions. The questionnaire was, in part, based on the questionnaire developed within the spring 2020 survey in the German National Cohort (kindly provided by the GNC) [11] and also utilized in the AugUR study [12]. This enables comparability of TiKoCo questionnaire results with GNC results for those aged 20–69 years and with AugUR for those aged 79 years and older.

In the questionnaire, we asked for (i) COVID-19 related symptoms (cough, shortness of breath, respiratory problems, fever, chills, loss of smell/taste, bronchitis/pneumonia), other symptoms related to general infections (pain in extremities, diarrhea, nausea, red eye/eye infection, headache, fatigue, rhinitis); (ii) PCR-based SARS-CoV-2 testing (results, date, reason, symptoms at time of testing), hospitalization related to COVID-19, or generally to bronchitis/pneumonia since the start of the pandemic (as per 1 February 2020); (iii) household (living alone, ≥1 other person, nursing home); (iv) previous illnesses by self-report (heart disease, lung disease including asthma and chronic bronchitis, kidney disease, diabetes, hypertension, cancer, auto-immune disease, blood coagulation disorder); (v) education (highest level of school, university and/or vocational training) and employment (status, type of occupation); and (vi) lifestyle factors: smoking (status as current, former, never; number of cigarettes smoked per day); alcohol consumption (frequency of drinking, number of drinks typically consumed); TV consumption (days per week with TV consumption for >2 h); physical activity as categories of weekly hours of light activity (0, 0–2 h, 2–4 h, >4 h including bicycling, walking).

### 2.7. Blood Draw, Transport, and Antibody Measurements

Blood was drawn in a sitting position by qualified study personnel into a barcoded serum monovette (Sarstedt AG Co.KG, Nümbrecht, Germany). All tubes sampled per day were processed immediately. One aliquot was sent to University Hospital Erlangen for assessment in the YHLO SARS CoV-2 test (chemoluminescent immune assay, CLIA). Two tests, an in-house ELISA and Elecsys, Roche Diagnostics (Roche-Cobas, CLIA), were performed at the University Hospital Regensburg.

The Elecsys Anti-SARS-CoV-2 test (Roche Diagnostics GmbH, Penzberg, Germany) detecting nucleoprotein-(N)-directed complete Ig was operated on the Cobas pro e 801 module and the YHLO SARS CoV-2 test (Shenzhen Yhlo Biotech Co. Ltd., Shenzen, China) detects IgG antibodies to the N- and S-protein on the iFlash 1800 according to the manufacturers recommendations, respectively. A validated in-house ELISA detecting IgG antibody responses to the receptor binding domain (RBD) of the SARS-CoV-2 spike protein was performed essentially as described earlier [13]. Sensitivities and specificities provided by the manufacturers in the instructions for use are 99.5% and 99.8% (Roche Diagnostics), 97.3% and 96.3% (YHLO), respectively. For the in house ELISA, a sensitivity of 96% and specificity of 99.3% has been determined [13]. 

### 2.8. Latent Class Analysis to Derive the True Seropositivity Status and Seroprevalence in the Study Population

Latent class analysis (LCA), a classical modeling approach for discrete data developed more 70 years ago by Lazarsfeld [14], has increasingly been applied during recent decades in the context of infectious diseases when a number of different diagnostic tests but no established gold standard are available [15]. In the TiCoKo study, information from three different antibody tests could be used to derive the true seropositivity status based on the pattern of results from the antibody tests. In general, LCA identifies a set of discrete, mutually exclusive latent (i.e., unobserved) classes based on the observed pattern of a set of categorical variables. The basic idea of LCA in our setting is to treat the unobservable true seropositivity status as being equivalent to two latent classes (seropositive vs. seronegative) and to relate the observed antibody test results from the three tests to it via a statistical model. The critical model assumption in LCA is that the three antibody tests are independent conditional on the true seropositivity status, which is called ‘local independence’ in LCA. The validity of this assumption can be tested with different methods; we used the log-odds ratio check proposed by Garrett and Zeger [15]. Given that the model derived from LCA fits the observed data well, the method provides an objective way of classifying the contradictory pattern of results from antibody testing. We checked the goodness of fit of the model derived from LCA by comparing observed and model-expected frequencies of response patterns from antibody testing and calculated standard goodness of fit measures in LCA like BIC and CAIC. 

Application of LCA to our data yielded a statistical prediction of the true seropositivity status for each study subject. This allows, in a subsequent step, the estimation of seroprevalence in the study population and its subgroups defined by gender, age, and residence in local municipalities of the county of Tirschenreuth. Point estimates of seroprevalence were always accompanied by asymptotic 95%-confidence intervals (CI), computed by Wilson’s method, to provide the information of their precision. The statistical software SAS version 9.4 (SAS Institute Inc., Cary, NC, USA) was used for these computations, and LCA modeling was performed by employing a SAS extension [16].

### 2.9. Standardization to Derive Seroprevalence, the Factor of Underreported Infections and Infection Fatality Ratio in Study Capture Area

The statistical analysis of seropositivity data in our study sample yielded crude results on seroprevalence and associated measure like the factor of underreported infections and infection fatality ratios applicable to the study cohort only. Since our study sample represents a random sample of residents of Tirschenreuth county aged 14 years and older, we were able to compute standardized figures of these quantities that allow for a population-oriented interpretation for Tirschenreuth county. We used official information from the administrative county office of Tirschenreuth and the 26 local residents’ registration offices in the county of Tirschenreuth to derive its age and sex distribution as well as the distribution of the population over the municipalities of the county. By weighting and combining the age-, sex-, and the municipality-specific figures estimated from the study sample with their corresponding weights in the Tirschenreuth population, we were able to derive seroprevalences for all three factors (in situations when we report subgroup-specific results, we standardized for the two factors not defining the subgroup).

Accordingly, standardized seroprevalences were used as the basis to calculate the factor of underreported infections as well as the infection fatality ratio (IFR). The number of reported cases (individuals tested PCR positive for SARS-CoV-2 infection) and the number of COVID-19 related deaths were obtained from the local health authorities. Acknowledging the possibility that a fraction of individuals may be unable to develop or may have already lost SARS-CoV-2 specific antibodies at the time point of the blood draw [2,17,18,19], we may have slightly underestimated the factor of underreported infections and overestimated the calculated IFR, respectively (see discussion).

When addressing the effect of residence in senior care homes, we divided the municipalities of the county into two subgroups: one group of municipalities with at least one senior care home and the remaining group of municipalities without any senior care home. In a further step, we also calculated standardized seroprevalence, the factor of underreported infections, and IFR for the individuals of all those above 70 years of age not residing in a senior care home and for individuals across all age groups not residing in a senior care home (in this additional analysis, the 13 study participants residing in a senior care home were excluded when computing crude seroprevalences and the 920 inhabitants residing in senior care home in the county of Tirschenreuth were excluded when computing the weights for the standardization under the assumption that their age distribution resembled the age distribution of senior care inhabitants in Germany [20].

All 95%-CIs for standardized seroprevalences were computed using the Wilson’s method for binomial proportions assuming the weights as fixed constants. For factors of underreported infections and IFRs, simple CIs for binomial proportions ignore the uncertainty in officially reported infection and death counts. Therefore, we do not report such intervals for these figures; instead, we computed Bayesian credibility intervals accounting for these uncertainties in the following manner: for the upper and lower bound of the interval, we used the empirical 2.5% and 97.5% quantiles of 100,000 samples drawn from a beta distribution with parameters dependent on the observed counts of infections and deaths, respectively, where in each of the 100,000 samples, the seroprevalence incorporated in the calculation was sampled from a normal distribution with mean and standard deviation given by the standardized estimates in the corresponding study (sub-)group. This approach follows the method applied by Streeck et al. [7].

### 2.10. Statistical Models to Investigate the Association of Seropositivity with Potential Risk Factors

We used logistic regression to investigate the association of seropositivity (binary, LCA-derived) with potential risk factors. We considered the following person-specific covariates: (1) person characteristics age group (14–19, 20–69, 70+) and sex; (2) personal background as education in years (6–10, 11–15, 16–23), household size (1, 2, 3–5, 6+) and occupation during spring 2020 (grocery store, medicine, other); and (3) lifestyle factors as physical activity (high, low), alcohol consumption (drinks per day: 0, 0–0.25, 0.25–1, >1), body-mass-index (<18.5, 18.5–25, 25–30, >30), and smoking (current, ex, never). To further investigate the association of smoking with seropositivity, we performed additional analyses on the dose-response association of seropositivity with the number of smoked cigarettes (linear or non-linear, in the overall sample and current smokers, adjusted for age and sex). The associations were presented as odds ratios with 95%-profile likelihood confidence intervals. Associations with a *p*-value < 0.05 were considered as significant

### 2.11. Comparison of Reported Symptoms between Persons with and without SARS-Cov-2 Infection

We analyzed the self-reported information on the occurrence of COVID-19 related symptoms since the beginning of the pandemic by comparing the fraction of individuals reporting the specific symptoms between three groups: seronegatives, seropositives with self-reported positive PCR test (i.e., individuals aware of their infection), and seropositives without positive PCR test (individuals unaware of infection). Furthermore, we quantified the association of symptoms with seropositivity based on the odds ratios of symptom occurrence and seropositivity combining individuals with and without positive PCR test result in the seropositive group.

## 3. Results

### 3.1. Participant Characteristics and Response

Of the Tirschenreuth inhabitants ≥14 years of age, 6608 individuals were randomly selected in a sex-, and municipality-stratified fashion and invited to participate in the TiKoCo study in June 2020 and 6540 individuals obtained an invitation letter. Of the contactable individuals, 4203 individuals participated, yielding an overall net response of 64.26% (Figure 1). The response was higher among the age group 20–74 compared to those 14–19 years or 75+ years of age (Appendix A) and also differed amongst the 26 local municipalities (Appendix A). The 4203 participants included 48.3% men, age ranged from 14 to 102 years (Table 1). A total of 20.4% reported to be current smokers and median body-mass-index was 27.1 kg/m^2^ and 26.0 kg/m^2^ among male and female participants, respectively. When focusing on the 633 participants aged 70+, which is the age group most at risk for severe COVID-19 and COVID-19 related death, 7.2% were current smokers smoking on average 11 cigarettes per day and median BMI was 27.5 kg/m^2^, 1.9% (*n* = 15) lived in a nursing home. When comparing our participants aged 70+ to the population-based AugUR study of participants all aged 70+ by design, we found no major differences in demographic or lifestyle factors, except for a higher proportion of current smokers and a higher number of daily smoked cigarettes (AugUR 4.8% current smokers, average of six cigarettes per day).

### 3.2. Self-Reported Positive PCR-Based Test for SARS-CoV-2

Of the 4201 participants, 518 (12.33%) reported to have had been previously tested by PCR for SARS-CoV-2 when filling out the questionnaire including 74 reporting a positive PCR test result (1.76% of participants, Table 2). Of the 74 individuals with a reported previous positive PCR result, 66 were registered at the local health authorities with a positive PCR test for 1.57% of the study participants. The remaining eight individuals were not registered as being tested by local health authorities (misinterpretation of question, or test outside the county).

Among the 64,643 residents of Tirschenreuth county aged ≥14 years (numbers provided by local municipalities of the Tirschenreuth county), health authorities reported 1.71% registered SARS-CoV-2 cases until June 2 (*n* = 1108). This is similar to the observed fraction of study participants reporting a positive test result, which supports the notion that individuals with known infection were as likely to participate as those without known infection.

Seroprevalence for SARS-CoV-2 was assessed by two commercial (Roche Diagnostics and Shenzhen YHLO Biotech) and one in-house assay [13]. Roche’s Cobas Elecsys CLIA test detects nucleoprotein-(N)-directed complete Ig levels including IgG, the YHLO CLIA test used detects IgG antibodies to the N- and S-protein, and the in-house ELISA quantifies IgGs binding to the receptor-binding domain (RBD) of the spike protein. Seropositivity was detected in 8.38% (95%-CI 7.85–9.26; *n* = 4187), 8.93% (95%-CI 8.11–9.14; *n* = 4186), or 9.17% (95%-CI 8.33–10.08; *n* = 4200) for the YHLO, the Roche-Cobas, and the in-house ELISA test, among the individuals with a successful test result, respectively. Differences in the detected proportions might be attributed to (i) different specificities and sensitivities of the assays; (ii) different antigen-specific antibody levels; and/or (iii) different isotype distributions depending on the time since infection.

### 3.3. Latent Class Analysis to Estimate True Seropositives in Tikoco Study Cohort

LCA to define seropositivity for all 4201 study participants with at least one result from any of the three antibody tests resulted in 363 individuals who scored SARS-CoV2 antibody positive (8.64%). This LCA-based analysis of the true antibody status on the basis of the three antibody tests was possible as the local independence assumption of the LCA holds (Garrett and Zeger’s log odds ratio test for a violation of the assumption yielded *p*-values for three pairwise comparisons of antibody tests of 0.98, 0.87 and 0.81, respectively). In addition, the goodness of fit of the model resulting from LCA indicated an adequate match of our observed data of antibody response patterns to the model-expected frequencies derived by LCA (Appendix A).

### 3.4. Seroprevalence, Underreported Infections Factor and Infection Fatality Ratio in the County of Tirschenreuth

Taking into consideration the minor differences in the composition of our study sample compared to the Tirschenreuth population, the crude seroprevalences obtained for our study sample were standardized according to gender, age, and local municipality to the county population, yielding a standardized overall seroprevalence of 8.57% (*n* = 4201, 95%-CI 7.77–9.45), which was similar between women (*n* = 189, 8.64%; 95%-CI 7.54–9.89) and men (8.50%; 95%-CI 7.37–9.78) (Figure 2a, Appendix A). 

By 2 June 2020, health authorities registered 1108 PCR positive COVID-19 cases (1.71%) and 138 COVID-19 related deaths in the Tirschenreuth population aged 14 years and older (64,643 inhabitants ≥14 years). Based on the standardized seroprevalence of 8.57%, 4432 of 5540 individuals calculated to be seropositive in the district of Tirschenreuth had not been registered as a COVID-19 case based on a positive PCR test.

Acknowledging that a certain fraction of tested individuals may have failed to develop or may have already lost SARS-CoV-2 specific antibodies until blood draw [2,17,18,19], this indicated that at least 80.00% of infections had remained undetected by the massive PCR testing as performed in spring and early summer 2020 in this particular county (Figure 2b, Appendix A). This corresponds to an underestimation of the cumulative number of infections by a factor of at least 5.00 in the ≥14-year-old population. Due to higher proportion of PCR+ women (1.98%) versus men (1.47%), the factor of underreported infections differed between women (factor 4.35) and men (factor 5.92).

Of note, of the 66 registered PCR-positive study participants (74 according to self-report), we found four (5 of 74 self-reported) without antibodies (6.06% and 6.76%, respectively) (Table 2). This could be due to a false positive PCR test result, leading to a slight overestimation of the factor of underreported infections. Alternatively, this discrepancy could also be due to a primary failure of the four to raise antibodies after infection, or a secondary loss of antibodies between seroconversion and blood draw end of June/early July, 2020. In that case, the proportion of infected would be higher (9.1%) than the seroprevalence (8.57%). 

In light of the 138 people who died from or with COVID-19 in the county until 2 June (all aged ≥14 years), we calculated the IFR as 2.49% (95%-CI 2.06–3.02) for the population of county Tirschenreuth aged 14 years and older, with an IFR of 2.18% (95%-CI 1.65–2.90) for women and 2.81% (95%-CI 2.17–3.67) for men (Figure 2c, Appendix A). When accounting for a potential 6% of non-seroconverters or individuals, who lost their antibodies, we would have had an overall IFR of 2.34% (women 2.05%; men 2.64%).

### 3.5. Seroprevalence, Factor of Underreported Infections, and Infection Fatality Ratio by Age Groups

We found gender- and municipality-standardized seroprevalences to vary from 5.14% (95%-CI 2.84–9.19) to 10.27% (95%-CI 4.56–21.87) by age groups (Figure 2d, Appendix A). When comparing the number of expected seropositives in the study population by age groups with the respective number of reported PCR positive individuals, the factor of underreported infections varied in the extremes from 12.15 (95%-CI 6.75–19.32) among teenagers (14–19 years) to 1.67 (95%-CI 1.00–3.12) in the elderly above 85 years, meaning that 91.75% and 40.12% of infections were unregistered in these two groups, respectively. For the other age groups, the factor of underreported infections varied between 3.04 to 7.21 (Figure 2e, Appendix A).

Based on the number of individuals estimated to be seropositive in the Tirschenreuth population by age group, we also determined the age-dependent IFR (Figure 2f, Appendix A). With the exception of one person among the 20- to 29-year-olds, no death related to COVID-19 was reported to health authorities for persons under 50 years of age. IFRs were low to moderate, <0.5% up to 59 years of age, and 0.98% (95%-CI 0.49–2.04) for those aged 60 to 69 years, but then dramatically increased with age exceeding 10% for individuals older than 75 years (95%-CI 5.84–30.21) and 30% (31.55%, 95%-CI 16.48–99.1%) for the elderly of 85 years and older.

### 3.6. Seroprevalence by Municipalities

In our study sample, seroprevalences differed markedly between municipalities (Figure 3a). Being aware of sparse numbers of inhabitants for some smaller municipalities, we nevertheless took an effort to standardize crude seroprevalence estimates to the study population by municipality, to better understand the regional infection dynamics in spring 2020. Standardized seroprevalences ranged from 22.65% (95%-CI 14.42–33.69) to 1.00% (95%-CI 0.16–5.47) (Figure 3a, Appendix A). Interestingly, we observed a gradient in seroprevalences from eastern (high) to western (low) municipalities, separated by a forest belt. In addition to geographic barriers, patterns of municipalities with high seroprevalence and others with moderate to low seroprevalence may be explained by a wide range of aspects: founding effects by ski-travelers returning from Italy or Austria, the local character of beer festivities prior to and during carnival season, proximity of public PCR test centers and/or family doctors as well as a growing awareness of infection occurrence followed by rigid testing.

Due to sparse numbers for some municipalities, we refrained from demonstrating factors of underreported infections and IFR per municipality. An important question was the extent to which the 62 COVID-19 related deaths observed in senior care homes by June 2 compared to a total of 138 COVID-19 related death cases in the complete county affected the IFR. Acknowledging previous reports on the high death toll in senior care homes, we separated municipalities into those with any senior care homes and those without (11 versus 15, respectively) (Figure 3a, Appendix A). For this, age- and sex-standardized seroprevalence was 8.90% (95%-CI 7.89–10.01) and 8.26% (95%-CI 6.93–9.83), respectively (Figure 3b). Considering the numbers of reported PCR positive cases (888 versus 219), this translated into a factor of underreported infections of 4.08 (95%-CI 3.83–4.99) and 8.48 (95%-CI 6.28–9.66) for the municipalities with and without senior care homes, respectively (Figure 3c). This marked difference may—in part—be explained by extensive PCR testing in senior care homes including social contacts of employees and senior citizen residents. When calculating the expected number of seropositives in these two groups of municipalities (3904 and 1725, respectively) and comparing it to the number of deaths (125 and 13, respectively), we found IFRs of 3.20 (95%-CI 2.60–3.97) and 0.75% (95%-CI 0.44–1.33), respectively (Figure 3d). Thus, this underscores the pivotal role of senior care homes on population-based estimates of IFRs.

### 3.7. Sensitivity Analysis Focusing on Individuals Not Living in Senior Care Homes

A large proportion of the observed COVID-19 related deaths were observed in senior care homes (*n* = 62 of the 124 deaths among individuals 70+ years of age, assuming senior care home residents are 70+). While individuals living in senior care homes (*n* = 920, 1.42% of the population ≥14 years) were part of our random drawing of invited participants to capture the full population, this is a very specific group of individuals in terms of comorbidities and living situation. The computation of interpretable IFRs for individuals living in nursing homes alone by our design is challenged by low response (*n* = 13, 12.15% of 107 invited senior care home residents compared to 53.84% participation among individuals aged 70+ not living in senior care homes and 57.45% among the overall population aged 70+).

To understand the impact of senior care homes on IFRs, we computed sex- and municipality standardized seroprevalences and IFR estimates for individuals aged 70+ not living in senior care homes and compared these to the estimates for all individuals aged 70+ (combining age groups 70–74, 75–79, 80–84, 85+ from the previous analysis). We observed similar seroprevalence in this restricted analysis for individuals aged 70+ not living in senior care homes (7.38%, 95%-CI 5.71–9.57) compared to all 70+ (7.79%, 95%-CI 6.06–10.03). The IFR for individuals 70+ not living in senior care homes was 7.54% (95%-CI 5.34–11.28; #deaths = 62), which was substantially lower than the IFR for those aged 70+ including senior care home residents (13.19%, 95%-CI 9.87–18.71, #deaths = 124).

We also evaluated how the total IFR across all age groups was affected when omitting individuals living in senior care homes: again, there was no impact on standardized seroprevalence across all ages (8.47, 95%-CI 7.67–9.33; compared to 8.57%, 95%-CI 7.77–9.45, see above). The IFR for individuals across all ages omitting those living in senior care homes was 1.41% compared to the 2.49% in the total study population derived above.

### 3.8. Reported Previous Illness

A “less healthy” elderly population in Tirschenreuth compared to other areas in Germany could explain some of the excess of COVID-19 deaths observed, particularly a less healthy elderly population. Thus, we asked participants about preexisting illness. We compared our participants aged 70+ with a reference study, the population-based AugUR study participants, all 70+ per design, from a geographically nearby region in Bavaria, Regensburg [12]. The questions for pre-existing illness were similar in the two studies, but AugUR had an in-person interview. For pre-existing illnesses such as obesity, cardiovascular disease, type 2 diabetes, and lung disease, the respective proportions in the two studies showed no major differences (Table 3); reports of cancer, chronic kidney disease, and hypertension were more common in AugUR compared to TiKoCo participants aged 70+, which may be partly explained by the different mode of administering the questions. Overall, there was no evidence for the TiKoCo participants to have been particularly predisposed to severe COVID-19 by pre-existing illnesses derived from self-reported height and weight. Relative (%) and absolute (#) frequencies based on information of [n] participants. Reports among individuals aged 70+ years (median age = 77.0 years, 25th–75th percentile 72.0–81.0) were compared with the AugUR study (median age = 78.9 years, 25th–75th percentile 75.7–82.5 years).

### 3.9. Reported Symptoms and Hospitalization

Among the 4162 individuals providing information on the experience of any COVID-19 related symptoms since the start of the pandemic (as per 1 February 2020), 3799 were negative for SARS-CoV-2 specific antibodies and 363 were seropositive (Figure 4). Interestingly, 12.9% (*n* = 47) of the seropositive individuals reported no symptom at all (i.e., asymptomatic infected), 82.4% (*n* = 299) reported at least one symptom, but did not report hospitalization due to COVID-19, and 4.7% (*n* = 17) reported hospitalization due to severe COVID-19. We found almost all symptoms more frequently reported among seropositive subjects compared to seronegatives, except eye inflammation and headache (highest odds Ratios, OR, for olfactory or taste problems and fever, OR = 21.94, 16.18, or 6.54, respectively) (Figure 4). These results support the role of most of these symptoms for SARS-CoV-2 infection, with a predominant role of olfactory and taste impairment. Reported symptoms in all participants indicate small differences by sex and age groups.

Among the unaware infected (i.e., participants with positive antibody status and without report of a positive SARS-CoV-2 PCR test), symptoms were reported less frequently than among participants with positive antibody status and report of positive SARS-CoV-2 test (Figure 4). One exception was the report of rhinitis, which was more frequent among the unaware infected than among the known infected and substantially more than among the antibody-negatives. This would be in line with an underestimation of rhinitis in its role of SARS-CoV-2 infection. 

Interestingly, symptoms were experienced clearly more frequently not only among the known infected, but also among the unaware infected when compared to the antibody-negatives (Figure 4). The fact that the unaware infected report symptoms less frequently than known infected is in line with the fact that asymptomatic or mild symptomatic infected were missed in the targeted testing (at least at this early stage of the pandemic in spring and early summer 2020).

We also found a strong association between individuals reporting a severe bronchitis/pneumonia (i.e., needing bed rest or physician or hospitalization) since the start of the pandemic (as per 1 February, 2020): the ORs for being antibody-positive among those reporting bronchitis/pneumonia related bed-rest or physician or hospitalization were 3.4, 3.2, or 15.4, respectively (Appendix A). Most of the antibody-positive individuals with bronchitis/pneumonia requiring hospitalization were aware of their infection (*n* = 11 among the 14 with hospitalization and antibody-positive). The majority of individuals reporting hospitalization for bronchitis/pneumonia since the start of the pandemic was infected (14 of the 24).

### 3.10. Association of Demographic and Lifestyle Factors with Seropositivity

We were interested in the question whether demographic characteristics were associated with higher seropositivity and thus higher probability of SARS-CoV-2 infection. We evaluated the association of age, sex, education years, size of household, or having worked in a profession with higher probability of contact (during the spring 2020 lockdown in Tirschenreuth, medical/nursing profession, and cashiers/salesperson in groceries) with seropositivity in a logistic regression model. We found no evidence for association with age, sex, education, or household size (*p*-Values > 0.05) (Figure 5, Appendix A). With regard to occupation, we found no increased seropositivity among cashiers/salespersons in groceries, but a doubling of the odds of being antibody-positive among medical and nursing professions compared to other occupations (OR = 2.26, 95%-CI: 1.53–3.28, Figure 5).

We were also interested whether seropositivity was associated with factors capturing a less healthy lifestyle (low physical activity, current or ex-smoking, drinking alcohol, or increased body-mass-index as a marker for excess calorie intake). We found no association with any lifestyle factor, except for current smoking. Remarkably, the odds of being seropositive was substantially decreased for current smokers compared to never-smokers (OR = 0.36, 95%-CI: 0.24–0.53), while there was no association for ex-smokers (Appendix A). This would correspond to a 2.8-fold increase in the odds of being seropositive among never-smokers or ex-smokers compared to current smokers (Figure 5).

The finding that current smoking was associated with a smaller probability of being seropositive was counter-intuitive under the hypothesis that smoking was associated with a less healthy attitude and more risky behavior. We thus conducted several sensitivity analyses in the search for potential confounding effects and questioning the possibility of whether an impaired ability to mount antibodies might explain our observations [21,22] (Supplementary Note): we found stable associations across age groups and sex (Appendix A) and a dose-response effect (Appendix A). The association persisted when comparing the known infected with the known uninfected (i.e., restricting to the *n* = 501 tested, OR = 0.35 for current smoking associated with positive versus negative PCR-test report), which is in line with a hypothesis that current smoking was associated with a lower risk of infection (Appendix A). This finding was not compromised by a higher proportion of current smokers among those tested, as previously reported by others [23].

A reverse epidemiology effect could induce a bias here, when individuals being current smokers at the height of the outbreak included severely ill individuals (e.g., cancer, severe heart disease) that prevented individuals from going outside or primed individuals to be particularly careful, avoiding infection. However, the smoking-associated severe diseases would more likely affect older individuals and are less likely among younger individuals; therefore, the stable effect estimates across age groups provide evidence against such bias that would fully explain the strong association across age groups. Another potential bias needs to be considered when smokers who were infected were less likely to participate in the study. However, most individuals have not known their seropositivity at the time of questionnaire completion and participation.

## 4. Discussion

Data on the seroprevalence for SARS-CoV-2 infections from population-based studies are superior to convenience sample data, but such data are limited in Germany. Our population-based study recruited 4203 individuals aged at least 14 years from the county hit the most and very early by the pandemic in Germany in spring 2020. We invited a random sample of Tirschenreuth county inhabitants and achieved a high response at 64%. We derived a standardized seroprevalence of SARS-CoV-2 of 8.6% with little variation across age groups, lower among current smokers and higher among individuals working in the medical or nursing profession. Age-specific IFRs were <0.5% for age groups below 60 years, 1.0% for those aged 60–69, 13.2% for those aged 70+, and overall IFR across age groups was 2.5%. We found a substantial impact by COVID-19 related deaths in senior care home residences, which comprised almost 46% of Tirschenreuth county deaths, reflected by an IFR of 7.5% among individuals aged 70+ and an overall IFR of 1.4% when excluding senior care home residence from the computation. 

The seroprevalence of 8.5% implied that only one out of five SARS-CoV-2 infections had been registered by health authorities by community targeted testing at that time. While the seroprevalence and thus the implicated incidence of infection was stable across all studied age groups, the lower number of registered infections in the young yielded a factor of underreported infections as high as 12.1 in 14 to 19-year-olds compared to 1.7 in the ≥85-year-olds. This is consistent with asymptomatic forms of COVID-19 at a younger age and a testing strategy focusing on symptomatic individuals, particularly at this early phase of the pandemic in spring 2020. Our findings underscore the need for SARS-CoV-2 testing, particularly in the young and suggests a note of caution to deduce infection rates from registered COVID-19 cases by a factor of underreported infections that is not adjusted for age.

We found several interesting aspects with regard to symptoms, in line with previous studies (e.g., [11,12,24,25]), particularly when comparing individuals unaware of their infection to the aware. We found a doubling of seroprevalence among individuals having worked in the medical or nursing professions during the outbreak compared to other occupations, consistent with previous studies (e.g., [24]), which might reflect limited access to appropriate protective gear at that very early stage. We did not find higher seroprevalence in individuals working in groceries. Particularly important to discuss is the finding that current smoking compared to never smoking, but not ex-smoking, was associated with a decreased risk of being seropositive and for reported infection. The association was strong, stable across age groups and sex, and exhibited a dose-response-effect. While the sample size of smokers among individuals with reported PCR-test was limited, the stable risk estimates and the few individuals with reported infection not showing antibodies suggests that the finding was not due to a lack of antibody building, but due to lower infection probability. This does not allow for the conclusion that smoking per se was protective. While smoking is generally linked to a less healthy and more risky behavior, a behavior associated with smoking that guards against infection could result in the same observation (e.g., gathering socially more outside or less frequently). A lower seroprevalence among smokers was also found by other population-based studies [24,25]. A biological hypothesis for a protective effect of nicotine has been suggested previously [26,27]. This was based on the observation of a lower prevalence of smokers among individuals hospitalized due to COVID-19. However, individuals affected with severe COVID-19 requiring hospitalization were predominantly older individuals and there were fewer smokers among older individuals (7.2% in our study among those aged 70+, versus 23.6% among those aged 20–69), rendering age as a confounder in these analyses. Further functional studies are warranted to understand the reason for this observed lower seroprevalence and the lower PCR-test reported infections among smokers. Any conclusion that smoking was preventive for SARS-Cov-2 infection needs to be rigorously challenged and, if substantiated, conclusions made need to weigh in the adverse public health impact due to the severe other implications of smoking like lung cancer.

Our study was unique by employing three different test systems, two commercial assay platforms, and one in-house assay to determine SARS-CoV-2 specific antibody responses. These assay systems varied (i) regarding the viral target for the specific antibodies (S-RBD for the in-house assay, S and N protein for the YHLO assay, and N protein only for the Roche assay); (ii) regarding the technical setup (in-house ELISA versus 2 commercial CLIAs); and (iii) regarding the detected antibody isotypes (IgG versus total Ig including IgA and IgM). Furthermore, all three assay systems varied according to the manufacturers’ specifications in terms of clinical sensitivity and specificity. Not unexpectedly, the crude seropositivity varied among the three platforms from 8.38% to 9.17% in the study participants. In order to determine the true seropositivity, we employed LCA, a statistical approach that has repeatedly been used in the past to resolve discrepancies between diagnostic tests in the absence of an established gold standard [28]. The statistical model resulting from LCA showed an excellent fit to the observed pattern of results from the three antibody tests and fulfills the crucial assumption of local independence necessary to draw valid conclusions from the model. The model-derived definition of seropositivity uses the available information from all three antibody tests, combines it in an objective manner, and constitutes the basis for our computation of seroprevalences and associated measures like underreported infections and IFR.

Based on registered COVID-19 related deaths obtained from health authorities and the number of Tirschenreuth inhabitants with infection during spring 2020 projected from standardized seroprevalence, we derived IFR estimates. As expected, IFR estimates substantially depended on age, resulting in nearly zero up to age 59 and 1.0%, 4.2%, 11.0%, 9.3%, and 31.5% for our age groups of 60–69, 70–74, 75–79, 80–84, or 85+, respectively. While it is difficult to compare overall IFRs across published studies due to different age ranges included and different age distributions, a recent meta-analysis derived a meta-regression model using age-specific IFRs across 27 population-based studies worldwide [29]. Based on that model, predicted IFR were <0.5% for ages below 60 and 1.4%, 3.4%, 6.1%, 11.3%, or 21.9% at ages 65, 72.5, 77.5, 82.5, and 88 years, respectively, reflecting the midpoints of our age groups. This is in line with our findings taking the confidence intervals into account.

We would like to note that we used the estimated seroprevalence as the proportion of individuals that have been infected at any time between the start of the pandemic in February 2020 and the time of the blood draw in June/July 2020. This neglects a small fraction of infected individuals that do not build antibodies or have already lost them [2,17,18,19]. While this would lead to underestimating the proportion of infected and overestimate IFRs, this is the same line of conduct as in most other studies (e.g., in the Levin et al., meta-analysis). More data are required to better understand non-seroconverting and antibody decline over time to allow for solid accounting for this aspect in IFR estimates.

The overall IFR at 2.49% was high compared to the reported IFR of 0.35% for Gangelt, another German hot spot early during the pandemic [7]. The difference in IFR may be, at least in part, explained by a smaller proportion of elderly inhabitants in the municipality of Gangelt compared to that in Tirschenreuth county (9.5% versus 12.0% aged 75+) [30,31]. Our data suggest that another important determinant for the extent of IFR, besides pure age, is the inclusion of senior home residents. While 13.7% of the registered COVID-19 infections in Tirschenreuth and 46% of COVID-19 related deaths [9] were residents of senior care homes, only 7.6% of the registered COVID-19 cases were senior care home residents in Gangelt [7,9]. When separating the Tirschenreuth communities into those with and those without any senior care homes, we found a total IFR of 3.2% and 0.8%, respectively. Importantly, the IFR for individuals aged 70+ of 13.2% reduced to 7.5% when excluding senior care home residents from the deaths and the seropositives. The deaths among senior care home residents also impacted the total IFR across all age groups: when excluding senior care home deaths and respective seropositives, the total IFR was 1.4% rather than 2.5% originally. This underscores the dominant impact of senior care home residents in the computation of IFRs.

Concern has been raised for population-based studies that include senior care home residents to underestimate seroprevalence due to low response and thus to overestimate COVID-19 IFRs [10]. However, excluding senior care home residents clearly underestimates total IFR. Recruiting the elderly, particularly senior care home residents, is a substantial challenge. For this reason, many population-based seroprevalence studies excluded individuals above the age of 70 years [32] or excluded senior care home residents by design (e.g., [33,34]) or by analysis [24]. Nevertheless, senior care home residents are a part of our populations. In our study, we included senior care home residents into our random sampling and provided mobile study teams to visit seniors in private homes as well as senior care homes when appropriate. By this, we limited non-response in the elderly, but still observed a lower response compared to all. We would like to point out that, despite all efforts (i.e., home visits), the response among senior care home residents aged 70+ with 13 participants (12.15%) was limited compared to those aged 70+ not living in senior care homes (57.45%) and the overall participation of the 70+ population (53.48%). This was the reason why we did not provide IFRs restricted to senior care home residents. Nevertheless, inclusion of this most affected group with sensitivity analysis excluding them appeared to be a fair attempt to help understand COVID-19 related fatality in the 70+ population.

It was hypothesized that the high Tirschenreuth death toll was potentially, at least in part, related to more frequent comorbidities or a less healthy lifestyle in this population [8,9]. However, self-reported lifestyle factors and comorbidities among the Tirschenreuth study participants aged 70+ (i.e. the age group where most COVID-19 related deaths occurred) compared well with self-report from individuals 70+ from another population-based study in a city in the vicinity of Tirschenreuth, Regensburg [12] did not provide evidence for this hypothesis. The only observation where these study participants aged 70+ differed was a higher proportion of individuals aged 70+ who were current smokers and more cigarettes were smoked in Tirschenreuth compared to Regensburg and the number of cigarettes smoked per day was also higher on average. However, it is not plausible that this can fully explain the high death toll by COVID-19 in Tirschenreuth. A better explanation might be infections in senior care homes: The majority of deaths of the hard-hit European countries at that time happened in senior care homes [35] and a large proportion of deaths in the U.S. also followed this pattern [36]. As suggested by our data, the high death toll in the county of Tirschenreuth and the relatively high IFR can, at least in part, be attributed to outbreaks in senior care homes, particularly at this early stage of the pandemic in Europe. At the time of the outbreak, awareness of the extent of the problem as well as protective gear like masks for the community had been limited. Still, outbreaks in senior care homes continue to happen with devastating outcome, but will hopefully be controlled by targeted vaccination efforts.

The scheduled follow-up investigations of study participants will allow for further analyses directed toward monitoring the temporal development of these SARS-CoV-2 related metrics such as seroprevalence, underreported infections factor, and IFR in the study population as well as the durability of the antibody response to the SARS-CoV-2 infection. Our present results provide important and comprehensive information from a population-based epidemiological perspective that will help to understand the COVID-19 pandemic.

## 5. Conclusions

While age has been recognized as the predominant determinant of COVID-19 IFR, our data underscores and quantifies the role of SARS-CoV-2 infections of senior home residents. This large impact is under-recognized when investigations focus on super-spreading events affecting a limited region for a limited time rather than a larger region reflecting all aspects of the population including several senior care homes. This impact is also under-recognized when senior care homes are excluded by study design or by requiring study participants to visit a study center. We included the elderly and senior care home residents in our random study sample and supported their participation by home visits. 

Our study is the only German seroprevalence study covering a complete German county. While our results have important implications in understanding the differences and communalities in IFRs across studies internationally, it has highly relevant implications to understand the pandemic in Germany: the only existing IFR estimate in a harder hit German municipality yielded an IFR of 0.36%. An overall IFR of 2.5% including senior care home residents and 1.4% excluding them may reflect the extent of the problem when a full county is affected.

## Figures and Tables

**Figure 1 viruses-13-01118-f001:**
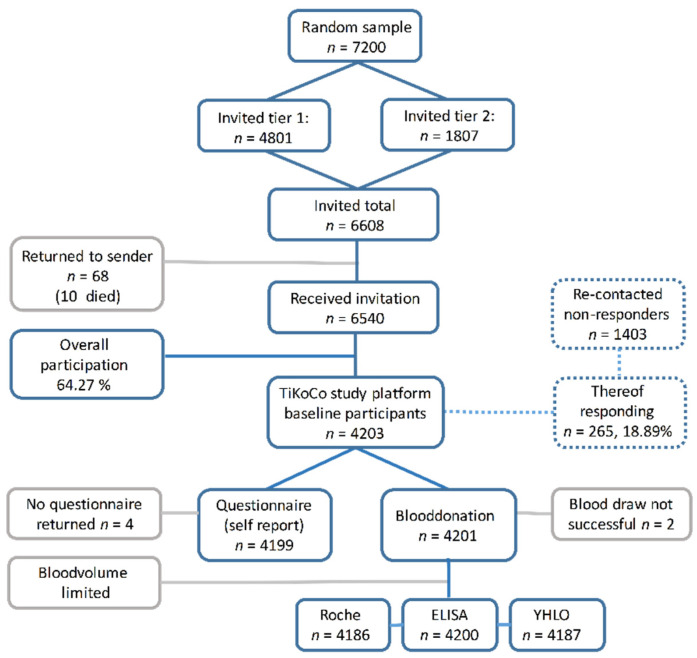
Summary of the TiCoKo study design. The strategy and numbers (*n*) underlying the random sampling and recruitment of study participants, the collection of information via a self-reporting questionnaire as well as the testing strategy to determine the true serostatus of the participants via three independent test formats (Roche-Cobas, in-house ELISA and YHLO) are shown. Measures to minimize and understand non-response are indicated (dashed line), reasons for drop-outs of participants, numbers of returned questionnaires, and successful blood samples are given (grey line). Figure was designed using PowerPoint 365 for Windows, Microsoft, Redmond Washington, DC, USA, www.microsoft.com.

**Figure 2 viruses-13-01118-f002:**
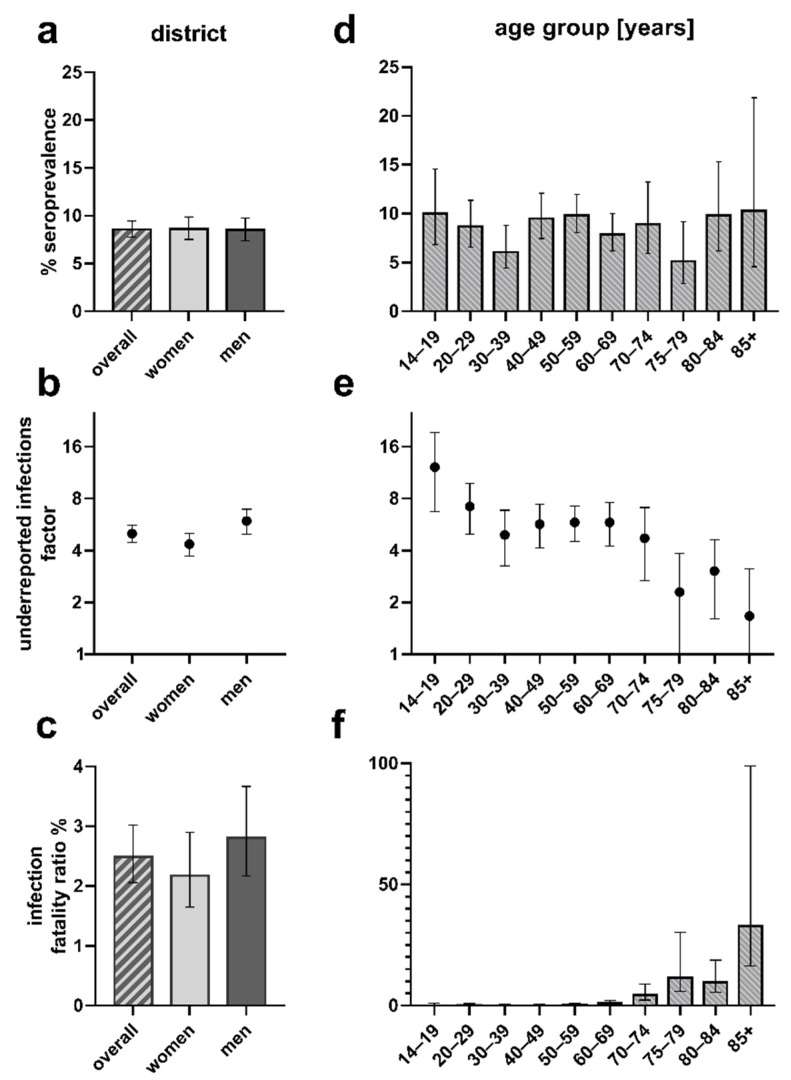
Seroprevalence, factor of underreported infections, and infection fatality ratios overall and by age groups. Standardized seroprevalence (%), dark figure factor and infection fatality ratio (%) in the overall county population (**a**–**c**) and the indicated age groups (**d**–**f**). Error bars represent 95% confidence intervals (95%-CI), respectively. Figure was designed using GraphPad Prism version 8.4.3 for Windows, GraphPad Software, La Jolla, CA, USA, www.graphpad.com.

**Figure 3 viruses-13-01118-f003:**
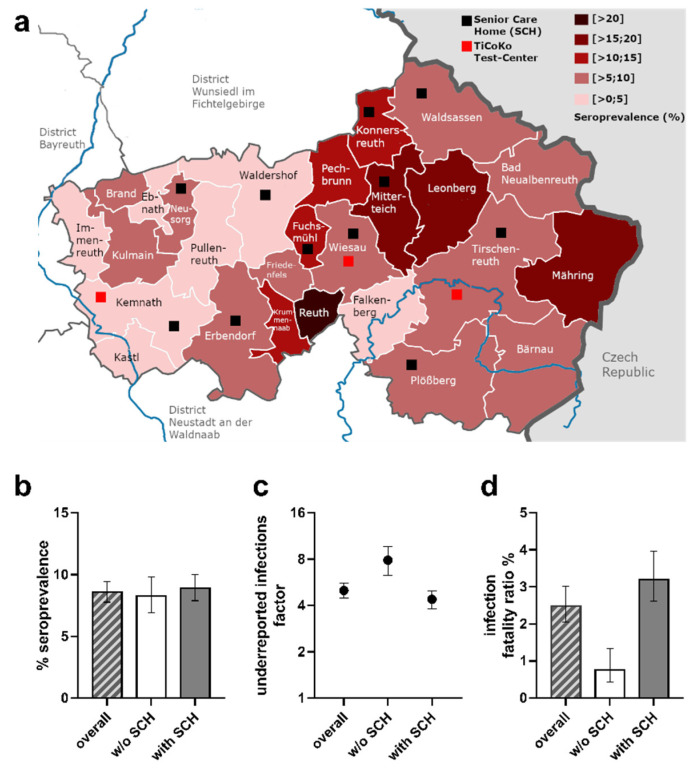
Seroprevalence, factor of underreported infections, and infection fatality ratios by municipalities with and without senior care homes. (**a**) Standardized seroprevalence (%) determined for inhabitants of the local municipalities in county Tirschenreuth. (**b**–**d**) Standardized seroprevalence (**b**), factor of underreported infections (**c**) and infection fatality ratio (**d**) in the overall county population, in the population of local municipalities without senior care homes (*w/o* SCH) and with senior care homes (with SCH). The 95% confidence intervals (95%-CI) are indicated, respectively. Red squares indicate the location of the study test centers, black squares highlight the municipality association of senior care homes. Figure was designed using GraphPad Prism version 8.4.3 for Windows, GraphPad Software, La Jolla, CA, USA, www.graphpad.com, and GIMP 2.10.22 for Windows, The GIMP Development Team, California, USA, www.gimp.org.

**Figure 4 viruses-13-01118-f004:**
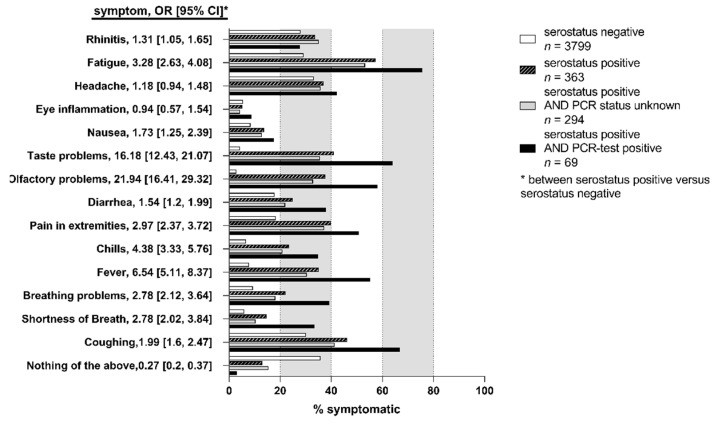
Symptoms. Percent individuals of the indicated categories, who developed one or several of the indicated symptoms. Odds ratios, OR, as well as 95%-CI [CI] testing serostatus positive versus serostatus negative (unadjusted) are depicted. Figure was designed using GraphPad Prism version 8.4.3 for Windows, GraphPad Software, La Jolla, CA, USA, www.graphpad.com.

**Figure 5 viruses-13-01118-f005:**
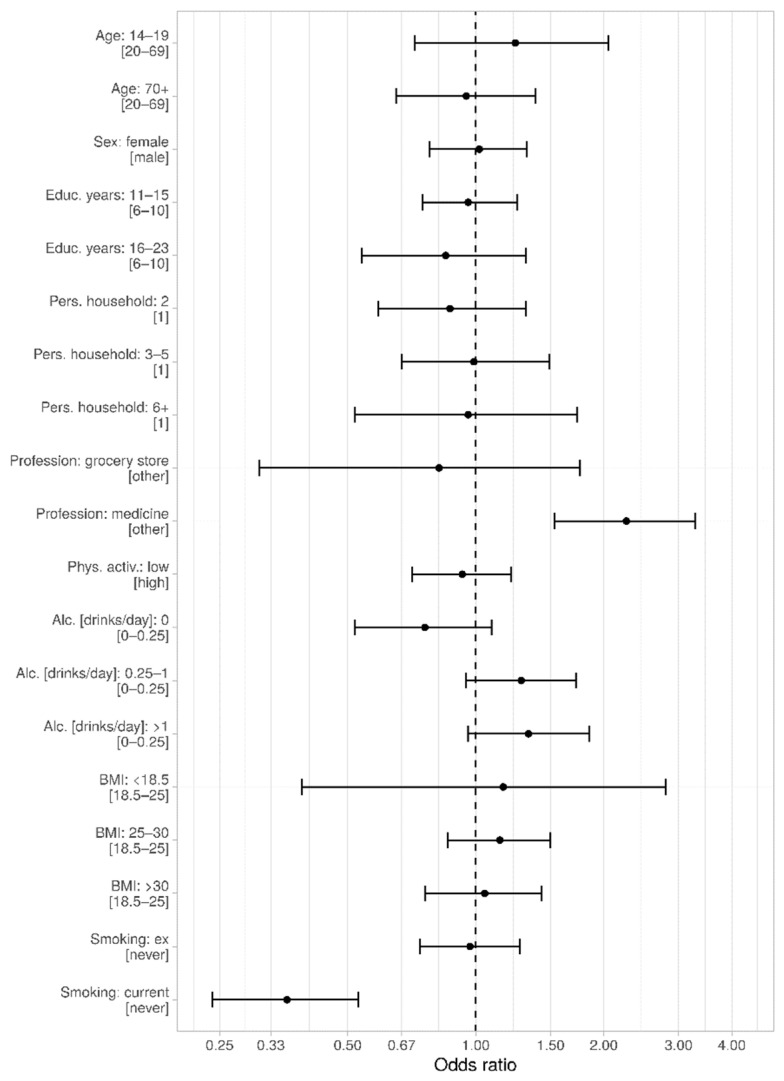
Association of demographic and life style factors with seropositivity. Odds ratios including the 95%-CI are indicated, respectively. *p* < 0.001 (***). Analysis was conducted in R (R Core Team (2020). R: A language and environment for statistical computing. R Foundation for Statistical Computing, Vienna, Austria. URL http://www.R-project.org/). This figure was produced using the package ggplot2 (Wickham, H. (2009) ggplot2: elegant graphics for data analysis. Springer New York).

**Table 1 viruses-13-01118-t001:** Participant characteristics. Shown are characteristics of the 4203 study participants (age, sex) and other characteristics for the 4201 study participants who filled out the questionnaire. Shown are median and (interquartile-range, IQR) or relative (%) and absolute (#) frequencies compared to the non-missing [number of individuals of the category].

	All	Me*n*	Wome*n*	Age 14–19	Age 20–69	Age ≥ 70
Demographic Factors						
Age, Med. (IQR)	52.0 (35.0, 64.0) [*n* = 4203]	51.5 (35.0, 63.0) [*n* = 2032]	52.0 (35.0, 64.0) [*n* = 2171]	17.0 (15.0, 18.0) [*n* = 227]	49.0 (35.0, 59.0) [*n* = 3343]	77.0 (72.0, 81.0) [*n* = 633]
Men, % (#)	48.3 (2032) [*n* = 4203]	100.0 (2032) [*n* = 2032]	0.0 (0) [*n* = 2171]	47.6 (108) [*n* = 227]	48.4 (1619) [*n* = 3343]	48.2 (305) [*n* = 633]
Educations years ^1^, Med. (IQR)	11.0 (10.0, 14.0) [*n* = 4104]	11.0 (10.0, 14.0) [*n* = 1986]	11.0 (10.0, 12.0) [*n* = 2118]	10.0 (6.0, 11.0) [*n* = 222]	11.0 (10.0, 14.0) [*n* = 3285]	10.0 (8.0, 11.0) [*n* = 597]
Life Style Factors						
Height [cm], Med. (IQR)	171.0 (165.0, 178.0) [*n* = 4179]	178.0 (174.0, 183.0) [*n* = 2019]	165.0 (161.0, 170.0) [*n* = 2160]	172.0 (165.0, 179.0) [*n* = 226]	172.0 (165.0, 179.0) [*n* = 3328]	168.0 (161.0, 174.0) [*n* = 625]
Smoking (curr.), % (#)	20.4 (853) [*n* = 4178]	21.9 (442) [*n* = 2022]	19.1 (411) [*n* = 2156]	10.6 (24) [*n* = 227]	23.6 (784) [*n* = 3328]	7.2 (45) [*n* = 623]
Smoking (ex), % (#)	24.6 (1028) [*n* = 4178]	29.4 (594) [*n* = 2022]	20.1 (434) [*n* = 2156]	3.1 (7) [*n* = 227]	25.0 (833) [*n* = 3328]	30.2 (188) [*n* = 623]
Num. cig daily ^2^, Med. (IQR)	15.0 (10.0, 20.0) [*n* = 788]	15.0 (10.0, 20.0) [*n* = 394]	10.0 (8.0, 15.0) [*n* = 394]	10.0 (1.0, 11.5) [*n* = 14]	15.0 (10.0, 20.0) [*n* = 736]	11.0 (10.0, 18.8) [*n* = 38]
Num. drinks daily ^3^, Med. (IQR)	0.6 (0.2, 1.2) [*n* = 3305]	0.6 (0.2, 1.5) [*n* = 1723]	0.2 (0.0, 0.6) [*n* = 1582]	0.2 (0.1, 0.6) [*n* = 143]	0.6 (0.2, 1.2) [*n* = 2749]	0.6 (0.2, 1.5) [*n* = 413]
Members Household						
Living alone (1), % (#)	12.7 (522) [*n* = 4120]	11.4 (228) [*n* = 1994]	13.8 (294) [*n* = 2126]	0.4 (1) [*n* = 223]	9.9 (328) [*n* = 3302]	32.3 (193) [*n* = 598]
2, % (#)	34.9 (1438) [*n* = 4120]	34.8 (693) [*n* = 1994]	35.0 (745) [*n* = 2126]	2.7 (6) [*n* = 223]	34.0 (1123) [*n* = 3302]	51.7 (309) [*n* = 598]
3–5, % (#)	47.0 (1932) [*n* = 4120]	48.3 (962) [*n* = 1994]	45.6 (970) [*n* = 2126]	83.8 (187) [*n* = 223]	50.3 (1664) [*n* = 3302]	13.6 (81) [*n* = 598]
>5, % (#)	5.5 (228) [*n* = 4120]	5.6 (111) [*n* = 1994]	5.5 (117) [*n* = 2126]	13.0 (29) [*n* = 223]	5.7 (187) [*n* = 3302]	2.5 (15) [*n* = 598]
Senior care home ^4^, % (#)	0.3 (13) [*n* = 4203]	0.05 (1) [*n* = 2032]	0.6 (12) [*n* = 2171]	0.0 (0) [*n* = 227]	0.0 (0) [*n* = 3343]	2.1 (13) [*n* = 633]

^1^ Education years were computed as number of years at school plus years at vocational schools and/or university including doctoral time, if applicable. ^2^ Number of cigarettes smoked daily are provided for current smokers. ^3^ Number of alcoholic drinks for individuals drinking any alcohol (computed from the frequency of drinking and number of drinks when drinking, with a drink being equivalent to 0.33 l beer, 0.125 l wine, or 4 cl hard liquor). ^4^ Residence in senior care homes was not assessed via questionnaire but from checking the addresses of all study participants.

**Table 2 viruses-13-01118-t002:** PCR-test self-report and confirmed positive PCR vs. serostatus.

PCR Test Status	Serostatus Negative % (#)	Serostatus Positive % (#)
No test, *n* = 3683	93.32 (3437)	6.68 (246)
Test negative, *n* = 427	89.00 (380)	11.01 (47)
Test positive, *n* = 74	6.76 (5)	93.24 (69)
Test positive, confirmed, *n* = 66 *	6.06 (4)	93.94 (62)
Test status unknown, *n* = 17	94.12 (16)	5.88 (1)
Overall, *n* = 4201	91.36 (3838)	8.64 (363)

* A total of 66 out of 74 self-reported PCR-test positive individuals were confirmed by local health authorities.

**Table 3 viruses-13-01118-t003:** Previous illnesses reported by participants. Participants were asked whether they ever had a physician diagnosing any of the following diseases, except for obesity, which was derived from self-reported height and weight. Relative (%) and absolute (#) frequencies based on information of [n] participants. Reports among individuals aged 70+ years (median age=77.0 years, 25th–75th percentile 72.0–81.0) were compared with AugUR study (median age = 78.9 years, 25th–75th percentile 75.7–82.5 years).

	TiKoCo Study Participants						AugUR Study Participants ^a^
	All	Male	Female	Age 14–19	Age 20–69	Age ≥ 70	Age ≥ 70
Cancer, % (#)	5.0 (204) [*n* = 4100]	4.6 (90) [*n* = 1977]	5.4 (114) [*n* = 2123]	0.0 (0) [*n* = 222]	3.7 (120) [*n* = 3258]	13.5 (84) [*n* = 620]	27.6 (672) [*n* = 2439]
Kidney Dis., % (#)	3.4 (138) [*n* = 4100]	3.7 (74) [*n* = 1977]	3.0 (64) [*n* = 2123]	0.5 (1) [*n* = 222]	2.7 (89) [*n* = 3258]	7.7 (48) [*n* = 620]	25.6 (624) [*n* = 2439]
Obesity, % (#)	27.0 (1122) [*n* = 4154]	28.0 (562) [*n* = 2010]	26.1 (560) [*n* = 2144]	7.5 (17) [*n* = 226]	27.6 (913) [*n* = 3306]	30.9 (192) [*n* = 622]	30.2 (734) [*n* = 2432] ^b^
Cardiovasc. Dis., % (#)	9.9 (404) [*n* = 4100]	12.0 (238) [*n* = 1977]	7.8 (166) [*n* = 2123]	2.3 (5) [*n* = 222]	7.0 (228) [*n* = 3258]	27.6 (171) [*n* = 620]	28.0 (677) [*n* = 2420]
Type-2 diabetes, % (#)	7.7 (314) [*n* = 4100]	8.7 (172) [*n* = 1977]	6.7 (142) [*n* = 2123]	0.5 (1) [*n* = 222]	5.7 (185) [*n* = 3258]	20.6 (128) [*n* = 620]	21.4 (522) [*n* = 2440]
Lung diseases, % (#)	10.5 (432) [*n* = 4100]	10.6 (209) [*n* = 1977]	10.5 (223) [*n* = 2123]	9.5 (21) [*n* = 222]	10.0 (325) [*n* = 3258]	13.9 (86) [*n* = 620]	12.4 (302) [*n* = 2434] ^c^
Hypertension, % (#)	29.9 (1227) [*n* = 4100]	33.3 (659) [*n* = 1977]	26.8 (568) [*n* = 2123]	1.4 (3) [*n* = 222]	26.1 (850) [*n* = 3258]	60.3 (374) [*n* = 620]	70.6 (1723) [*n* = 2440]
Blood clotting, % (#)	2.2 (91) [*n* = 4100]	2.0 (39) [*n* = 1977]	2.4 (52) [*n* = 2123]	0.9 (2) [*n* = 222]	1.6 (52) [*n* = 3258]	6.0 (37) [*n* = 620]	n.a.
Autoimmune dis., % (#)	7.1 (291) [*n* = 4100]	4.1 (81) [*n* = 1977]	9.9 (210) [*n* = 2123]	1.4 (3) [*n* = 222]	7.1 (232) [*n* = 3258]	9.0 (56) [*n* = 620]	n.a.

^a^ AugUR (self-reported information at last visit): mean age 80.04 ± 5.11 (median 79.39, range 70.4–98.3); 52.1% female; ^b^ measured at study center; BMI >30 kg/m^2^; ^c^ Chronic bronchitis or asthma; n.a. not analyzed within AugUR.

## Data Availability

All authors declare that data and materials will be made available according to the guidelines of the journal.

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
