# Peer review of "Estimates and Determinants of SARS-Cov-2 Seroprevalence and Infection Fatality Ratio Using Latent Class Analysis: The Population-Based Tirschenreuth Study in the Hardest-Hit German County in Spring 2020"

_viruses, 2021, doi:10.3390/v13061118_

Round 1

Reviewer 1 Report

Ralf Wagner et al present a study on the seroprevalence of a German county in connection with the calculation of two indicators accounting for the epidemic of SARS-CoV-2, the IFR and the CFR. The article is interesting and very comprehensive, reflecting a very large mass of data. 1. On the other hand, there is a lack of clarity in particular on the definition of the two key indicators, IFR and CFR. Could you clearly explain in the article how you calculate these indicators? 2. The use of LCA is interesting in this context, it would have deserved a better justification with regard to more traditional classification methods. 3. The LCA likelihood ratio fit values are difficult to interpret since it is not known whether the true model is one of the hypotheses tested. Specify the assumptions in the Methods section and qualify your remarks line 416. A BIC of 59.7 does not represent a "high accuracy" in any way. 4. Another strong hypothesis of your article concerns the assertion lines 435-436: all infected produce antibodies. This is absolutely false and the literature is full of articles on the subject. Explain the consequences of this assumption on your results and discuss them absolutely. Minor: l. 429, "similar", give the p-value

Author Response

Reviewer #1

Ralf Wagner et al present a study on the seroprevalence of a German county in connection with the calculation of two indicators accounting for the epidemic of SARS-CoV-2, the IFR and the CFR. The article is interesting and very comprehensive, reflecting a very large mass of data.

--

Comment 1. On the other hand, there is a lack of clarity in particular on the definition of the two key indicators, IFR and CFR. Could you clearly explain in the article how you calculate these indicators?

Reply:  The reviewer is correct, this was not fully clear. We have re-worded particularly the first paragraph in the introduction and now state specifically a definition of CFR and IFR. We also added a paragraph to the methods section to better explain how IFR was calculated.

Revision:

The revised start of the introduction now reads:

Page 2, line 50, clean version: “COVID-19 case numbers reported to health authorities based on PCR testing continue to rise world-wide, but the precise cumulative number of infected individuals remains unknown. PCR testing frequencies and strategies vary largely between countries and over time, thus limiting the strength of the conclusions that can be drawn based on case fatality ratios (ratio of SARS-CoV-2 related deaths to the number of PCR positive cases reported to health authorities, CFR) [1]. Determining the number of infected individuals, the ratio of underreported SARS-CoV-2 infections, and the ratio of COVID-19 related deaths to infected (infection-fatality-ratio, IFR) helps to understand the extent of undetected infections (factor of underreported infections), to determine the level of herd immunity and to instruct public health measures (…)”  

Under Materials and Methods we did now supplemented the chapter “Standardization to derive seroprevalence, factor of underreported infections and infection fatality ratio in study capture area“, by adding an additional paragraph.

Page 7, line 310 clean version “ (…) Accordingly, standardized seroprevalences were used as basis to calculate the factor of underreported infections as well as the infection fatality ratio (IFR). The number of reported cases (individuals tested PCR positive for SARS-CoV-2 infection) and the number of COVID-19 related deaths were obtained from the local health authorities. Acknowledging the possibility that a fraction of individuals may be unable to develop or may have already lost SARS-CoV-2 specific antibodies at the time point of the blood draw [2,17–19], we may slightly underestimate the factor of underreported infections  and overestimate the calculated IFR, respectively (see discussion).”

--

Comment 2. The use of LCA is interesting in this context, it would have deserved a better justification with regard to more traditional classification methods.

Reply: We thank the reviewer for his/her remark. The use of LCA in diagnostic studies when no gold standard is available and information from several tests has to be combined to define the ‘true’ status is neither new nor uncommon. In a systematic review by van Smeden et al. (see reference no 15 in our manuscript) covering publications until November 2011 the authors identified 111 studies using LCA in such a situation. The authors found that the use of LCA in such studies increased sharply in the past decade prior to 2011, notably in the domain of infectious diseases. This trend persisted also after 2011. Therefore, we felt that a detailed justification for using LCA in our study is not necessary. In the revised version we have now, however, expanded the motivation for using LCA in our study.

Revision: The rephrased paragraph now reads:

Page 6, line 267, clean version: “Latent class analysis (LCA), a classical modelling approach for discrete data developed more 70 years ago by Lazarsfeld [14], has increasingly been applied during recent decades in the context of infectious diseases when a number of different diagnostic tests but no established gold standard are available [15], In the TiCoKo study information from three different antibody tests could be used to derive the true seropositivity status based on the pattern of results from the antibody tests. In general, LCA identifies a set of discrete, mutually exclusive latent (i.e. unobserved) classes based on the observed pattern of a set of categorical variables. The basic idea of LCA in our setting is to treat the unobservable true seropositivity status as being equivalent to two latent classes (seropositive vs. seronegative) and to relate the observed antibody test results from the three tests to it via a statistical model. The critical model assumption in LCA is that the three antibody tests are independent conditional on the true seropositivity status, which is called ‘local independence’ in LCA. The validity of this assumption can be tested with different methods; we used the log-odds ratio check proposed by Garrett and Zeger [15]. Given the model derived from LCA fits the observed data well, the method provides an objective way of classifying contradictory pattern of results from antibody testing. We checked the goodness of fit of the model derived from LCA by comparing observed and model-expected frequencies of response patterns from antibody testing and calculated standard goodness of fit measures in LCA like BIC and CAIC”.

--

Comment 3. The LCA likelihood ratio fit values are difficult to interpret since it is not known whether the true model is one of the hypotheses tested. Specify the assumptions in the Methods section and qualify your remarks line 416. A BIC of 59.7 does not represent a "high accuracy" in any way.

Reply. We agree that any interpretation of absolute values of goodness-of-fit (GOF) statistics obtained during the process of model diagnostics is problematic. In LCA model diagnostics the focus typically lies on two aspects: (i) the evaluation of the assumption of local independence between model variables that is crucial for the validity of the LCA approach and (ii) the comparison between observed and model-predicted frequencies for the response patterns of the model that is typically summarized in GOF measures based on the log-likelihood of the LCA model incorporating different penalization terms for more complex models.

We employed Garrett and Zeger’s log odds ratio test to address (i). We found no evidence for a violation of the crucial assumption of local independence. P-values of 0.98, 0.87 and 0.81, respectively, for three pairwise comparisons of model variables indicate that our data are compatible with the assumption of local independence and yield no concern regarding the validity of the LCA modeling approach at all. Considering (ii) the BIC (Bayesian information criterion by Schwartz) and the related measure CAIC by Bozdogan are the most often used GOF measures in LCA diagnostics, although other GOF measures like AIC and AICC are also advocated as useful. The strength of these measures in LCA diagnostics comes into play when several models of different complexity have to be compared and a decision about the final model has to be made. In our situation we were not confronted with such a model selection problem as model variables to be considered and the number of latent classes were given by the design of the study and the nature of the problem, respectively. In our case values for BIC and CAIC were thus only given for descriptive purpose in the original version of the manuscript. Which level of model accuracy the actual values reflect is a matter of subjective interpretation. Therefore, in the revised version of the manuscript we do not give the absolute values for BIC and CAIC accompanied by a qualifying interpretation in the results section any longer. Instead we incorporated a supplemental table 2 giving the complete information about the fit of LCA model in terms of a tabular comparison of observed and model-predicted frequencies for all data patterns of the LCA model.

Revised version. We slightly rephrased the description in the results section as follows:

Page 12, line 425, clean version: “(…) LCA to define seropositivity for all 4201 study participants with at least one result from any of the three antibody tests resulted in 363 individuals who scored SARS-CoV2 antibody positive (8·64%). This LCA-based analysis of the true antibody status on the basis of the three antibody tests was possible as the local independence assumption of the LCA holds (Garrett and Zeger’s log odds ratio test for a violation of the assumption yielded p-values for three pairwise comparisons of antibody tests of 0·98, 0·87 and 0·81, respectively). In addition, the goodness of fit of the model resulting from LCA indicated an adequate match of our observed data of antibody response patterns to the model-expected frequencies derived by LCA (Supplemental Table 2)”.

--

Comment 4. Another strong hypothesis of your article concerns the assertion lines 435-436: all infected produce antibodies. This is absolutely false and the literature is full of articles on the subject. Explain the consequences of this assumption on your results and discuss them absolutely.

Reply. We understand the reviewer’s argument and completely agree that there is - depending on the respective study - a discrepancy between reported cases and seroconversion. This was the reason why we carefully phrased the sentence in our original manuscript (line 435, 436) with "(...) Under the assumption that all ever infected produced antibodies detectable by our assessment, this indicated that 80·00% of infections had remained undetected by the massive PCR testing as performed in spring and early summer 2020 in this particular county (...)"

We also stated in the immediately following paragraph (original manuscript) that also our study observed a discrepancy between reported (PCR+) cases and antibody status (line 443-447): "(...) Of note, of the 66 registered PCR-positive study participants (74 according to selfreport), we found 4 (5 of 74) without antibodies (6·06% and 6·76%, respectively) (Table 2). This could be due to a false positive PCR test result, a primary failure to raise antibodies after infection, or a secondary loss of antibodies between seroconversion and blood draw end of June/early July, 2020 (...)".

We agree with the reviewer that the simplifying assumption that really 100% of infected present anti-bodies after 6 months is unrealistic. Still, the studies referenced in this manuscript - e.g. in the Levin et al. meta-analysis - mostly calculated IFRs based on seroprevalence studies without further accounting for these non-converters or individuals who already lost antibodies. Nevertheless, it is interesting to consider a certain proportion of individuals without antibody-building or loosing them and to understand the impact of this on IFRs. When using the proportion of individuals with known PCR-positive test results for whom we did not detect antibodies (4 out of 66; 6.06%) as an estimate for the proportion of individuals without antibody building/losing, the proportion of infected overall increased from 8.57% to 9.09% and the overall IFR of 2.49% would decrease to 2.34%.

However, it is unclear how good this estimate of 6% of non-converter/antibody-decline based on these small numbers is in general and in dependency by age. Studies on antibody decline and non-converter are just at a start and will provide the opportunity to improve these estimates in the future. Recent data, however, suggest, that our observation is in good agreement with large studies reporting seroconversion rates ranging from 91 to 99% within the first weeks following infection (Wajnberg et al. 2020; doi: 10.1016/S2666-5247(20)30120-8; Gudbjartsson et al., 2020; doi: 10.1056/NEJMoa2026116 ) and a moderate decay of antibody levels over time (Dan et al., 2021 doi: 10.1126/science.abf4063)  (Vanshylla et al., 2021; https://doi.org/10.1016/j.chom.2021.04.015).

Revised version: For the sake of clarity and in order to avoid any misunderstanding, we rephrased the above text passage in the “results section” in the revised version as follows:

Page 12, line 448, clean version: “(…) Acknowledging that a certain fraction of tested individuals may have failed to develop or may have already lost SARS-CoV-2 specific antibodies until blood draw, [2,17–19] this indicated that at least 80·00% of infections had remained undetected by the massive PCR testing as performed in spring and early summer 2020 in this particular county (Figure 2b, Supplemental Table 3). This corresponds to an underestimation of the cumulative number of infections by a factor of at least 5·00 in the ≥ 14-year-old population. Due to higher proportion of PCR+ women (1·98%) versus men (1·47%), the factor of underreported infections differed between women (factor 4·35) and men (factor 5·92).

Of note, of the 66 registered PCR-positive study participants (74 according to self-report), we found 4 (5 of 74) without antibodies (6·06% and 6·76%, respectively) (Table 2). This could be due to a false positive PCR test result leading to a slight overestimation of the factor of underreported infections. Alternatively, this discrepancy could also be due to a primary failure of the 4 to raise antibodies after infection, or a secondary loss of antibodies between seroconversion and blood draw end of June/early July, 2020. In that case, the proportion of infected would be higher (9.1%) than the seroprevalence (8.57%).

In view of the 138 people who have died from or with COVID-19 in the county until June 2nd (all aged ≥14 years), we calculated the IFR as 2·49% (95%-CI 2·06-3·02) for the population of the county Tirschenreuth aged 14 years and older, with an IFR of 2·18% (95%-CI 1·65-2·90) for women and 2·81% (95%-CI 2·17-3·67) for men (Figure 2c, Supplemental Table 3). When accounting for a potential 6% of non-seroconverters or individuals, who lost their antibodies, we would have had an overall IFR of 2.34% (women 2.05%; men 2.64%).”

We also added a clarifying paragraph to the discussion:

Page 23, line 760, clean version: “(…) Based on that model, predicted IFRs were < 0·5% for ages below 60 and 1·4%, 3·4%, 6·1%, 11·3%, or 21·9% at age 65, 72·5, 77·5, 82·5, and 88 years, respectively, reflecting the midpoints of our age groups. This is in line with our findings taking the confidence intervals into account. We would like to note that we used the estimated seroprevalence as the proportion of individuals that have been infected at any time between the start of the pandemic in February 2020 and the time of the blood draw in June/July 2020. This neglects a small fraction of infected individuals that do not build antibodies or have already lost them [2,17–19], While this would lead to underestimating the proportion of infected and overestimate IFRs, this is the same line of conduct as in most other studies in the Levin et al., meta-analysis. More data is required to better understand non-seroconverting and antibody decline over time to allow for solid accounting for this aspect in IFR estimates.”

Reviewer 2 Report

Authors evaluated SARS COV2 seroprevalence, the dark figure and the IFR in the county of Tirschenreuth, Germany. Authors also assessed demographic and lifestyle factors potentially associated with SARS COV2  seroprevalence. This epidemiological study covers a complete German county and results may have important implications to understand the pandemic.

The manuscript is well written and structured. Sections are clearly presented. Information presented in this paper are important for the scientific community working in the field.

I have few comments:

Methods

Lines 247-259: Authors should provide sensitivity and specifiticity for the serological tests used to determine SARS-COV-2 antibodies

Results: in general, I suggest to review this section and to shorten it

Author Response

Reviewer # 2

Authors evaluated SARS COV2 seroprevalence, the dark figure and the IFR in the county of Tirschenreuth, Germany. Authors also assessed demographic and lifestyle factors potentially associated with SARS COV2  seroprevalence. This epidemiological study covers a complete German county and results may have important implications to understand the pandemic.

The manuscript is well written and structured. Sections are clearly presented. Information presented in this paper are important for the scientific community working in the field.

--

Comment 1: Lines 247-259: Authors should provide sensitivity and specificity for the serological tests used to determine SARS-COV-2 antibodies

Reply: Sensitivities and specificities provided in the respective “instructions for use” and determined for the in house ELISA [13] are now given under material and methods. Due to the fact that the LCA was used to determine the true seroprevalence, a correction for test sensitivity / specificity is dispensable here.

Revision: page 6, line 260: (…) “Sensitivities and specificities provided in the manufacturers instructions for use are 99.5% and 99.8% (Roche), 97.3% and 96.3% (YHLO), respectively. For the in house ELISA, as sensitivity of 96% and specificity of 99.3% has been determined [13].“

--

Comment 2: Results: in general, I suggest to review this section and to shorten it

Reply:  We agree with the reviewer.

We have reduced content in Table 1, removed Table 3 completely and shifted Table 5 to Supplemental Tables. We also eliminated Figure 4a from the main text. We also condensed and removed supplementary tables: Tables 6 and 7 were eliminated, table 9 and 10 were integrated into one Table (new Table 9a and b). We also shortened and edited the results part throughout, and particularly in the section “previous illnesses” and the section about the sensitivity analyses on the smoking finding, which in part has been shifted to Supplemental notes (see mark-up version of the manuscript).

Reviewer 3 Report

This was a very interesting and well written manuscript. The methods and results were presented in great detail.

There are many tables and figures in this manuscript, and the results sections goes on for several pages. It might be helpful to remove some tables or figures, and to shorten the results section to focus on the key results you planned to present.

While I'm familiar with the term "dark figure", it seems to be not widely known outside of German language contexts. Is there another term you could use?

Table 1 might be easier to read if the columns for males and females were removed. 

Based on Table 1, it appears that only 12 subjects in the sample were in senior care homes. Is that correct? If so, the conclusions based on this subset are not very strong. This should be reflected in the text. Additionally, this would be a group of potential subjects who are probably among the least likely to participate.

Author Response

Reviewer # 3

This was a very interesting and well written manuscript. The methods and results were presented in great detail.

--

Comment 1: There are many tables and figures in this manuscript, and the results sections goes on for several pages. It might be helpful to remove some tables or figures, and to shorten the results section to focus on the key results you planned to present.

Reply:  We agree with the reviewer.

We have reduced content in Table 1, removed Table 3 completely and shifted Table 5 to Supplemental Tables. We also eliminated Figure 4a from the main text. We also condensed and removed supplementary tables: Tables 6 and 7 were eliminated, table 9 and 10 were integrated into one Table (new Table 9a and b). We also shortened and edited the results part throughout, and particularly in the section “previous illnesses” and the section about the sensitivity analyses on the smoking finding, which in part has been shifted to Supplemental notes (see mark-up version of the manuscript).-
--

Comment 2: While I'm familiar with the term "dark figure", it seems to be not widely known outside of German language contexts. Is there another term you could use?

Reply:  We consistently replaced throughout the manuscript, all figures, tables and the supplementary materials the term “dark figure” by “underreported infections”

--

Comment 3: Table 1 might be easier to read if the columns for males and females were removed. 

Reply:  We understand that our Table 1 was large. In epidemiological studies, there is usually a strong attention to how men and women are included in the study and how they differ. Thus, we would like to keep men and women. However, we felt that we can shorten in terms of rows (omitting the row on height and weight, as this is combined as body-mass-index, which is more relevant, and condensing the data given on household size). The Table 1 is now much smaller and indeed better to digest.

--

Comment 4: Based on Table 1, it appears that only 12 subjects in the sample were in senior care homes. Is that correct? If so, the conclusions based on this subset are not very strong. This should be reflected in the text. Additionally, this would be a group of potential subjects who are probably among the least likely to participate.

Reply. Upon revisiting Table 1, we realized that the number of subjects „Age >70“ (right column, bottom line) should be 13 instead of 12 (12 women, 1 man as indicated correctly in the 3 columns (left)). We apologize for this mistake, which is now corrected in Table 1 of the revised manuscript 

We fully agree with the reviewer. We therefore already added a note of caution in the results section of the original manuscript stating „(…) The computation of interpretable IFRs for individuals living in nursing homes alone by our design is challenged by low response (n=13, 12·15% of 107 invited residence inhabitants).“

Reply: we now extended this statement as follows, page 15 line 543, clean version „(…) The computation of interpretable IFRs for individuals living in nursing homes alone by our design is challenged by low response (n=13, 12·15% of 107 invited residence inhabitants compared to 53.84 among individuals aged 70+ not living in senior care homes or tot he 57.45% of the overall population aged 70+).“

We have extended the discussion to make this clearer:

Page 24, line 798, clean version „(…) In our study, we included senior care home residents into our random sampling and provided mobile study teams to visit seniors in private home as well as senior care homes when appropriate. By this, we limited non-response in the elderly, but still observed a lower response compared to all. We would like to point out that, despite all efforts (i.e. home visits), the response among senior care home residents aged 70+ with 13 participants (12.15%) was limited compared to those aged 70+ not living in senior care homes (57.45%) and the overall participation of the 70+ population (53.48%).  This was the reason why we did not provide IFRs restricted to senior care home residents. Nevertheless, inclusion of this most affected group with sensitivity analysis excluding them appeared to be a fair attempt to help understand COVID-19 related fatality in the 70+ population.“

Round 2

Reviewer 1 Report

No further comment